# Moving towards a core measures set for patient safety in perioperative care: An e-Delphi consensus study

J. P. Dinis-Teixeira[1]*, Ana Beatriz Nunes[1,2], Andreia Leite[1,2,3], Willemijn L. A. Schäfer[4], Claudia Valli[5,6], Ismael Martínez-Nicolas[7], Ayshe Seyfulayeva[1], Pedro Casaca Carvalho[1,2], Anna Rodríguez[5,6], Daniel Arnal-Velasco[7], Irene Leon[7], Carola Orrego[5,6,8], Paulo Sousa[1,2], on behalf of the SAFEST Consortium and the SAFEST Scientific Advisory Group[¶]

1 National School of Public Health, NOVA University Lisbon, Lisbon, Portugal, 2 NOVA National School of Public Health, Comprehensive Health Research Center, CHRC, NOVA University Lisbon, Lisbon, Portugal, 3 Department of Epidemiology, Instituto Nacional de Saúde Doutor Ricardo Jorge, Lisbon, Portugal, 4 Department of Surgery, Northwestern Quality Improvement, Research & Education in Surgery (NQUIRES), Northwestern University, Chicago, IL, United States of America, 5 Avedis Donabedian Research Institute, Barcelona, Spain, 6 Universidad Autónoma de Barcelona, Barcelona, Spain, 7 Spanish Anaesthesia and Reanimation Incident Reporting System (SENSAR), Alcorcon, Spain, 8 Network for Research on Chronicity, Primary Care and Health Promotion (RICAPPS), Barcelona, Spain

¶ Membership of the SAFEST Consortium and the SAFEST Scientific Advisory Group is provided in the Acknowledgments
* jpd.teixeira@ensp.unl.pt

**Data Availability Statement:** All relevant data are within the manuscript and its Supporting Information files.

## Abstract

A Core Measures Set (CMS) is an agreed standardized group of measures that should be assessed and reported in research for a specific condition or clinical area. This study undertook the development of a CMS for Patient Safety through a two-round, web-based Delphi consensus approach, in the context of the "Improving quality and patient SAFEty in surgical care through STandardisation and harmonization of perioperative care in Europe" (SAFEST) project—a collaborative, patient-centered and evidence-based European Union-funded project that aims to generate action-oriented evidence in perioperative care. We developed an Initial List of Measures via an umbrella review following the deployment of an e-Delphi method with an inclusive panel of experts to prioritize measures towards a consensualized Final List of Measures. All measures were rigorously assessed for both importance and feasibility. After the two rounds of the e-Delphi consensus method we observed 13 preoperative measures (40.6% of the initial number), 24 intraoperative measures (66.7%), 25 postoperative measures (20.3%) and 23 mixed period measures (41.1%) met consensus criteria for both importance and feasibility. Higher scores were detected in importance ratings compared to feasibility across all groups of measures. Importantly, numeric averages regarding pain-related measures differed in the assessment of patients when compared to that of Healthcare Professionals (HCPs). This work not only informs future SAFEST iterations but also sets a precedent for research into valid, patient-centered, and action-oriented perioperative safety measures.

**Funding:** The work underlying this manuscript is encompassed in the European project SAFEST (Improving quality and patient SAFEty in surgical care through STandardisation and harmonisation of perioperative care in Europe). This project receives funding from the European Union's Horizon Europe research and innovation programme under grant agreement No 101057825. Link: https://cordis.europa.eu/project/id/101057825 The funders had no role in study design, data collection and analysis, decision to publish, or preparation of the manuscript.

**Competing interests:** The authors have declared that no competing interests exist.

# Introduction

## Global context

The World Health Organization (WHO) defines patient safety as a framework of organized activities in health care that consistently and sustainably lower risks, reduce the occurrence of avoidable harm, make errors less likely and reduce the impact of harm when it occurs [1].

Since the publication of the "To Err is Human" report by the Institute of Medicine (IOM) in 2000 [2], much attention has been brought to the central importance of patient safety for the delivery of quality healthcare services.

In 2015, all 193 member states of the United Nations (UN) committed to engage in efforts to achieve Universal Health Coverage (UHC) [3]. As such, all individuals and respective communities should receive adequate health services without suffering financial hardship.

However, UHC should not be achieved at the expense of the foundational bioethics precept first do no harm, from which derives the principle of non-maleficence in Medicine [4]. Therefore, patient safety is considered paramount to the achievement of UHC.

## Prevalence and consequences of unsafe care

Adverse events from medical care are reported to amount to 42.7 million annually worldwide, causing the loss of 23 million disability-adjusted life years (DALYs) [5].

In addition to directly affecting the lives of patients and their carers, unsafe care bears a substantial economic impact. 1 in 10 US dollars spent on health care of high-income countries was spent in managing the consequences of safety errors [6]. Specifically for the European Union's (EU) public health sector the burden is also significant: patient safety-related cost accounts for around EUR 21 billion in direct costs yearly (roughly 1.5% of health expenditure).

Specifically for patients undergoing surgery, the mortality rate in Europe has been found to be higher than previously thought [7].

Some approaches tailored to tackle surgery-specific impacts of harmful care have already been implemented. The enforcement of a checklist based on the WHO Surgical Checklist effectively decreased mortality in patients undergoing non-day case surgery [8].

Although of critical importance, the accurate and reliable measurement of adverse events remains a significant challenge [9]. Importantly, the WHO urged governments to develop sets of "indicators for patient safety aligned with global patient safety targets" [1]. Measurement plays a critical role in patient safety by helping to identify adverse event trends, set patient safety priorities and thus support resource allocation decision-making, evaluating the effectiveness of patient safety interventions, and comparing safety performance across healthcare services.

## Core Measures Sets and the Delphi method

A Core Measures Set (CMS) is an agreed standardized group of measures that should be measured and reported in all future research or practice for a specific area [10].

Measures used in quality of health care and patient safety are usually classified as either a structure, process, or outcome (SPO) measure, as per the Donabedian conceptual model [11]. Importantly, the widely used concept of Core Outcomes Set (COS) falls under the wider scope of the CMS.

These sets do not imply that outcomes in a particular trial should be restricted to those in the set. Rather, there is an expectation that while the core measures will always be collected,

the researchers will continue exploring different ones [12]. The use of CMS might also bear an important impact on benchmarking for healthcare delivery services [13].

Multiple publications have reported the use of the Delphi method as a key pillar to the development of a CMS in different areas [14–17].

## The SAFEST project

"Improving quality and patient SAFEty in surgical care through STandardisation and harmonisation of perioperative care in Europe" (SAFEST) is a collaborative, patient-centered and evidence-based EU-funded project that aims to generate evidence within this field of knowledge [18].

As an intervention-oriented project with relevant expected impact in the quality and safety of provided perisurgical care, the scope of SAFEST is largely dependent on the measurement and evaluation of indicators. Specifically, the development of CMS serves two main objectives. Initially, it is expected to be used in the SAFEST study to oversee and assess safety outcomes in ten European hospitals where the SAFEST recommendations will be implemented. Additionally, it will play a role in upcoming European studies and initiatives focused on surgical safety. This will be achieved by promoting standardized outcome reporting in perioperative patient safety. Furthermore, the CMS will facilitate the comparison of data from various studies and inform efforts to expand the area of knowledge.

As part of this project, the development of a CMS for achieving an EU-wide consensus on relevant and feasible core measures to assess patient safety in perioperative care in surgical adult patients was planned.

The objective of this study is to contribute to a consensus among experts, ultimately resulting in the formulation of a consensualized list of measures which might work as a stepping-stone for assessing perisurgical care safety and quality in healthcare.

## Methods

An integrated approach was used to pursue the development of a CMS for Patient Safety in Perioperative Care. The methodology of this qualitative study based on a web-based Delphi survey (e-Delphi) comprised three steps. The Delphi method is a systematic method used for structuring the process of communication within a group of individuals. In medical sciences, Delphi methods might be employed as an alternative when conducting experimental or quasi-experimental studies are considered to be impractical due to ethical, pragmatic or economic constraints.

The method consists of a number of rounds of anonymous questionnaire surveys, in which experts are asked to provide their inputs on a specific topic. The responses are then analyzed, and feedback is provided to the experts for review. This process is usually repeated until a consensus is reached or until the level of agreement among the experts is satisfactory.

Importantly, studies that rely on the Delphi method can function as tools to further empower patients and patients' representatives by bringing them to the forefront of evidence generation [19–21].

Initially, a comprehensive list of measures for patient safety in perioperative care among surgical adult patients was created. This was mainly accomplished through an umbrella review of measures. Secondly, a group of healthcare professionals and patients' representatives was assembled in order to prioritize these measures using a two-round e-Delphi method.

Further iterations post-e-Delphi were considered so as to build upon the Final List of Measures (FLM) that arose from the consensus method.

This study has been registered in the COMET Database [22], curated by the Core Outcome Measures in Effectiveness Trials (COMET) Initiative.

## Consensus panel selection and recruitment

The research team aimed to ensure a diverse panel, with heterogenous expertise and backgrounds as well as balanced sociodemographic characteristics like gender.

Experts from key stakeholder groups including healthcare professionals, patients and patients' representatives, regulatory agency representatives, policymakers, private sector representatives, guideline developers, CMS methodologists/researchers, and governmental agency representatives were recruited during the first six months of the SAFEST project.

All experts were selected based on their experience, expertise in patient safety, perioperative care, or outcomes research. Additionally, experts needed to be adults and capable of completing the online surveys in English.

A pool of 67 experts was sent an invitation letter by e-mail with the following information: the SAFEST research team identification, funding of the study, a brief description of the e-Delphi and each of the two rounds, the expected time to be spent during the participation, projected timeline, information on the right to refuse or withdraw from the study, associated risks, benefits and compensation, as well as confidentiality information and contacts. All experts had the opportunity to contact the research team should they require additional information.

Ethical approval for SAFEST was given by IDIAP Jordi Gol's Research and Ethics Committee (CEIm code 22/146-P). For this study, all Delphi panel participants were required to provide written informed consent to partake in the eDelphi Modified Technique. The voluntary nature of participation was emphasized, and participants were explicitly informed about their right to withdraw from the study at any time.

## Development of the initial list of measures (ILM)

An umbrella review works under the assumption that multiple systematic reviews can be amalgamated into a cohesive, unified form [23]. In this case, thorough search was conducted across multiple databases: PubMed, EMBASE, Web of Science (Core Collection), Scopus, The Cochrane Library (Cochrane Database of Systematic Reviews), Cumulative Index to Nursing and Allied Health Literature, and COMET Initiative database.

In order to achieve the consolidated initial list, we added indicators retrieved from a secondary analysis from a systematic review of Guidelines and an umbrella review of non-clinical interventions to enhance perioperative patient safety encompassed into the SAFEST project [22].

All protocols for the reviews have been registered in PROSPERO. Work Package (WP) 6 umbrella review of outcomes and outcome measurement instruments for patient safety in perioperative care has been registered under CRD42022362921. WP2 systematic review has been registered under CRD42022347449 and WP4 umbrella review has been registered under CRD42023397419.

The measures initially identified underwent a refinement process that encompassed rewording, merging and deduplication. This was executed by the research team, resulting in a consolidated initial list of 247 measures.

## Achievement of consensus through an e-Delphi method

In the second stage of the process, a consensus panel experts' group was assembled in order to prioritize the initial list of measures based on the Delphi approach.

Although evidence on how to optimize patient engagement in research is still lacking, the existing body of knowledge is enough to underline the importance of pursuing such participation [24].

The e-Delphi is a modern approach aiming to digitize the Delphi method to enhance its effectiveness in coordinating collective and diverse group thoughts, while leveraging the aforementioned methodological benefits. By utilizing an Internet-based platform, the e-Delphi method streamlines communication between the researcher and the panel of experts, providing organization and control [25].

The e-Delphi used in this study involved two rounds and was conducted as an online survey on the Welphi® platform. The survey was distributed to the experts through personalized links automatically generated by the platform. Participants were able to complete the survey using any internet-connected device, such as mobile phones, laptops, or computers. To assist the consensus panel in this e-Delphi Technique, the researcher team developed a user manual. This manual provided guidance on the rationale and development of the CMS within the SAF-EST project, an overview of the e-Delphi methodology and structure, instructions for using the platform, and the procedures related to the e-Delphi method.

Participants were asked to rank each indicator regarding its perceived importance and feasibility of measurement (from now on, "feasibility"). A Likert 9-point scale was used, where the 1 meant the expert attributed no importance at all to the indicator or considered it to be extremely difficult to measure while value 9 was to be picked whenever the expert considered the indicator critical for inclusion or extremely easy to measure (Table 1).

The expert consensus process was conducted in two rounds between February and May 2023. Round 1 took place from February 24th to March 17th. Round 2 occurred between April 18th and May 2nd. Participants received summaries of key findings and the full list of measures, including descriptive statistics and anonymized comments at the conclusion of each round. The research team reviewed measures based on expert feedback and revised names, definitions, and descriptions between rounds. After Round 1, two new measures were added, 11 were merged, and four were removed based on expert suggestions. Reminders were issued to experts during each round.

Consensus criteria was set for both inclusion and exclusion across round 1 and round 2 of the e-Delphi. If an indicator was to be scored between 7 and 9 by 75% of the experts and between 1 and 3 by 15% or less of experts, it was considered to include. If an indicator was to be scored between 1 and 3 by 75% of the experts and between 7 and 9 by 15% or less of experts, it was considered to exclude.

Measures achieving consensus for inclusion according to the above criteria and without the need for rewording were included in the final list of measures.

**Table 1. Likert scale rating and corresponding descriptive values by importance and feasibility.**

| Value in Likert-scale | Importance | Feasibility |
|:---:|:---:|:---:|
| 1 | Not important at all | Extremely difficult to measure |
| 2 | Not important | Very difficult to measure |
| 3 | Not that important | Difficult to measure |
| 4 | Slightly important, but not critical | Measurable, but with important difficulties |
| 5 | Moderately important, but not critical | Measurable, but with difficulties |
| 6 | Important, but not critical | Measurable, but with minor difficulties |
| 7 | Very important | Easy to measure |
| 8 | Extremely important | Very easy to measure |
| 9 | Critical for inclusion | Extremely easy to measure |

Considering the consensus panel selection criteria used by the research team, a comparison between the results from HCPs and patients was set to be performed. This was to be achieved by analyzing the average rating attributed by patients of the top five highest rated measures by HCPs.

## Results

### e-Delphi round one findings

The first round of the e-Delphi survey was sent to 67 experts in patient safety, including 60 healthcare and research professionals and seven patients or patient representatives. In terms of gender balance, from the group of experts that started the e-Delphi, approximately 31 (55%) identified themselves as male. The dominant age group was 55–74 (n = 29; 51.8%), while the 34–54 age group was the second most prevalent (n = 29; 39.3%). Most experts had post-graduate education (n = 50; 89.3%).

From the initial pool of 67 experts contacted initially, 48 (72%) completed the survey, seven (10%) started but did not finish, and 12 (18%) did not begin. The eight incomplete e-Delphi responses were analyzed, handling missing data through complete-case analysis.

Round 1 respondents represented diverse expert categories. Most healthcare experts were doctors and nurses specialized in anesthesiology and surgery. The patients had surgery themselves or a direct family member did within the last 5 years.

Further detailed information on the profile of the experts can be found in S1 Table.

From the 247 initial consolidated list of measures included in Round 1, 40 (16.2%) reached a consensus to include regarding both importance and feasibility.

The distribution of the consensual round 1 measures by perioperative period and Donabedian's quality of care conceptual model Structure–Process–Outcome subgroups is displayed below in Table 2.

Of these 40 measures, only nine had no additional comments from experts in Round 1. From these nine, two were reworded based on comments on different measures. This resulted in eight measures being immediately endorsed to be integrated into the final list of measures. No measures reached a consensus to exclude. As such, no measures were excluded in this round and all those that did not reach consensus moved on to Round two of the e-Delphi.

The five highest rated measures in terms of importance on round one were "Equipment to administer oxygen to all patients undergoing procedures under sedation by anaesthesiologists is available", "Specialised equipment for the management of difficult airways is available where anaesthesia is given", "A preoperative up to date medication list is available in the medical records", and "Postoperative stroke (outcome)". Regarding feasibility, the five highest rated measures were "The operating time is recorded", "Length of surgery", "Intraoperative blood transfusion", "Defibrillators with cardiac pacing mode are available", and "Equipment for fluid and blood warming and rapid transfusion is available".

**Table 2. Distribution of measures that achieved round 1 consensus by perioperative period and SPO subgroup.**

| Perioperative period | Preoperative | | Intraoperative | | Postoperative | | Mixed | |
|---|---|---|---|---|---|---|---|---|
| S-P-O | n | % | n | % | n | % | n | % |
| Structure | 2 | 0.8 | 11 | 4.5 | 2 | 0.8 | 3 | 1.2 |
| Process | 2 | 0.8 | 6 | 2.4 | 0 | 0.0 | 1 | 0.4 |
| Outcome | 1 | 0.4 | 3 | 1.2 | 4 | 1.6 | 5 | 2.0 |

### e-Delphi round two findings

Out of the 55 experts who responded in the initial round, 49 participated in the second round, with 47 ultimately completing it. The general profile of the respondents of the second round very strongly overlapped with those from the first round in terms of gender, age group, highest level of education completed and role.

212 measures were evaluated according to its perceived importance and feasibility in round two. 15.6% (n = 33) of the measures reached consensus to include regarding both importance and feasibility, while none were consensually excluded.

From those 212 measures assessed in round two, higher scores were observed in importance compared to feasibility within both main classifications of perioperative period and Donabedian model category (Fig 1).

The five highest rated measures in terms of importance on round two were "Equipment to administer oxygen to all patients undergoing procedures under sedation by anaesthesiologists is available", "A preoperative up to date medication list is available in the clinical records", "There is an internal policy for resuscitation defined and diffused among professionals", "There is a well-defined internal protocol for major haemorrhage defined, including clinical laboratory and logistic responses, that is diffused among professionals", "There is a well-defined internal policy that ensures emergency drugs are available where anaesthesia is given and adequately stored defined and this policy is diffused among professionals". Regarding feasibility, the five highest rated measures were "There is a well-defined internal protocol for major haemorrhage defined, including clinical laboratory and logistic responses, that is diffused among professionals", "Fever", "A preoperative glucose monitoring is conducted in diabetic patients by a knowledgeable and trained professional based on the best available evidence", "Equipment to administer oxygen to all patients undergoing procedures under sedation by anesthesiologists is available", and "There is a well-defined internal policy that ensures emergency drugs are available where anaesthesia is given and adequately stored defined and this policy is diffused among professionals".

Of note, the two highest ranked importance measures from round two are similar to two of the highest ranked in round one. They are, however, not exactly the same as they have been reworded between rounds.

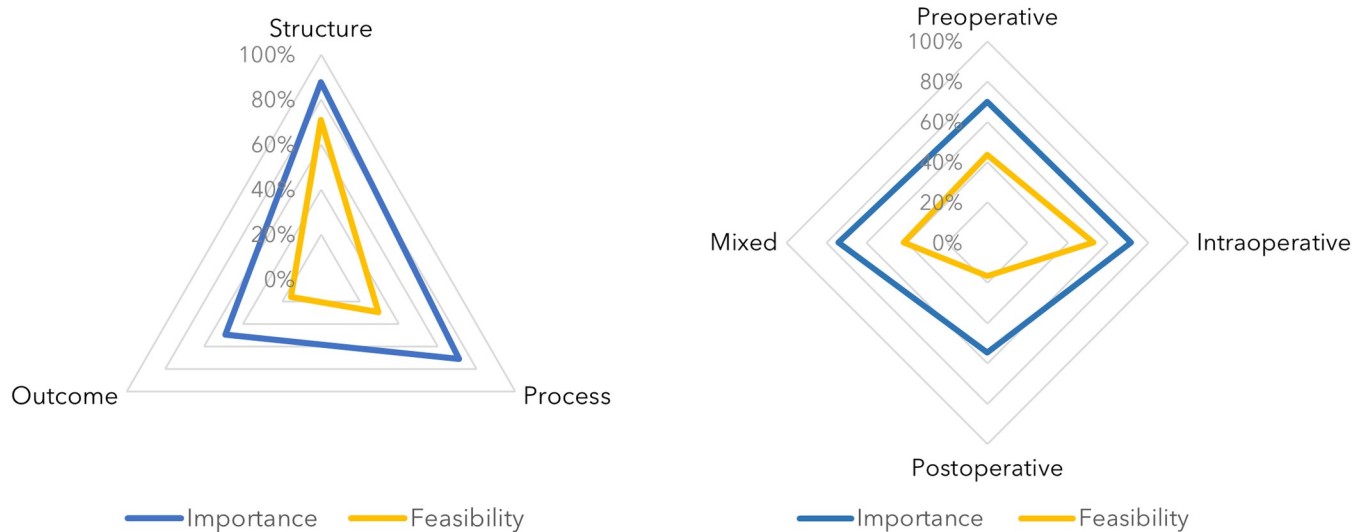

**Fig 1. Diagrams with the proportion of measures that reached consensus according to Donabedian category and perioperative period.**

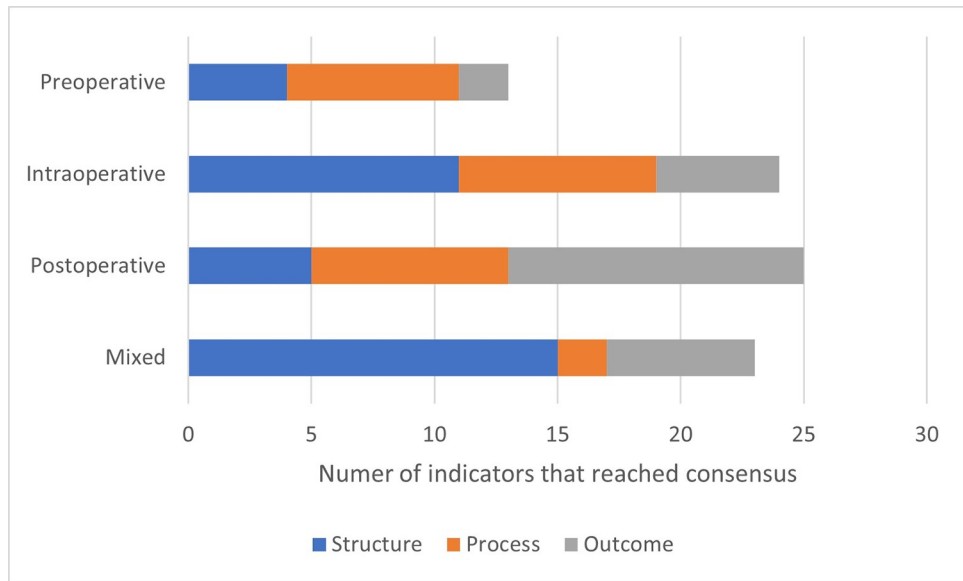

**Fig 2. Number of measures that reached consensus to include after both rounds of the e-Delphi Technique distributed by perioperative period and Donabedian model category.**

## Overall e-Delphi findings

Out of the initial list of measures from the two rounds, 34.1% (n = 85) achieved a consensus to be included on the FLM based on the ratings of importance and feasibility. 164 measures did not reach a consensus to include when considering both importance and feasibility simultaneously. No measures reached a consensus to exclude.

The number of overall agreed upon measures to include, categorized by perioperative period and Donabedian's quality of care model Structure-Process-Outcome subgroups, is shown in Fig 2.

Different percentage of agreement for consensus was reached across perioperative periods in the ILM. From the initial 32 preoperative measures, 36 intraoperative measures, 123 postoperative measures, 56 mixed period measures, the following achieved consensus after two rounds of the e-Delphi consensus method: 13 preoperative measures (40.6% of the initial number), 24 intraoperative measures (66.7%), 25 postoperative measures (20.3%) and 23 mixed period measures (41.1%).

The FLM can be found in S1 Appendix.

## Subgroup description: Patients and/or patient representatives and HCPs

By the end of the second round, the top five highest rated measures by HCPs on average in terms of importance were: "Equipment to administer oxygen to all patients undergoing procedures under sedation by anesthesiologists is available", "A preoperative up to date medication list is available in the clinical records", "There is an internal policy for resuscitation defined and diffused among professionals", "There is a well-defined internal protocol for major haemorrhage defined, including clinical laboratory and logistic responses, that is diffused among professionals", and "There is a well-defined internal policy that ensures emergency drugs are available where anaesthesia is given and adequately stored defined and this policy is diffused among professionals". The average rating given by patients or patients' representatives to each of these measures is shown in Table 3.

**Table 3. Average rating in terms of importance of the top 5 highest HCP-rated measures as graded by HCP and patients or patient representatives at the end of round two.**

| Indicator | HCPs average (standard deviation) | Patients average (standard deviation) |
|---|---|---|
| Equipment to administer oxygen to all patients undergoing procedures under sedation by anesthesiologists is available | 8.40 (0.80) | 8.60 (0.52) |
| A preoperative up to date medication list is available in the clinical records | 8.35 (0.85) | 8.60 (0.52) |
| There is an internal policy for resuscitation defined and diffused among professionals | 8.28 (0.97) | 8.40 (0.52) |
| There is a well-defined internal protocol for major haemorrhage defined, including clinical laboratory and logistic responses, that is diffused among professionals | 8.24 (0.79) | 8.80 (0.42) |
| There is a well-defined internal policy that ensures emergency drugs are available where anaesthesia is given and adequately stored defined and this policy is diffused among professionals | 8.13 (0.90) | 8.40 (0.85) |

Regarding patient-relevant outcomes, pain-related measures were assessed in this subgroup description of results. Although not specifically powered to assess significant differences among these two groups, numeric averages were higher as graded by patients or patients' representatives when compared with the assessment of HCPs regarding the same pain-related measures.

## Discussion

As patient safety gains attention as one of the key pillars of quality of care, governments, national and international organizations in healthcare, and research teams and consortiums like ours have taken steps towards addressing the burden of unsafe, harmful care.

De Vries et al. have clearly reported the relevance of developing comprehensive, multidisciplinary list of surgical safety measures in the reduction of surgical complications and mortality in hospitals [26]. This is in line with the dominant body of evidence regarding the importance of developing checklists and CMS for specific areas of care [8,27]. Building upon this idea, our work consisted of the design and operationalization of a two-round e-Delphi consensus method, through which we sought to discern the measures that garnered agreement among a heterogenous expert panel, encompassing healthcare professionals with diverse backgrounds and expertise. Elaborating on the results of an umbrella review (a systematic review of systematic reviews), the starting point was as broad as possible without compromising the expected compliance of experts in performing the required tasks.

One important limitation of using systematic reviews as the basis of a consensus work, as posed by Bampoe et al., is that this methodology only allows the identification and subsequent consensualization of existing clinical measures, rather than the development of novel ones [28]. This issue was addressed by allowing experts in our study to propose new measures during round one of the e-Delphi.

One other area of methodology worth discussing would be the number of Delphi rounds utilized. Regarding this issue, Erffmeyer et al. stated that, although some authors have considered otherwise, the use of two rounds might be insufficient to achieve stability in iterations [29]. Importantly, however, the definition of what constitutes a satisfactory level of agreement remains up for debate in the literature [30,31]. Moreover, it remains to be proven that the right number of rounds exist. An optimal approach to consensus in a Delphi might come from formally defining consensus criteria a priori [32].

In our study, after arriving at an ILM, a crucial decision to be made on this process regarding the consensus criteria to be used was pending.

The chosen method to probe for consensus in the present study revolves around the concept of 'controlled feedback'. This means experts were provided with a statistical analysis of the results from the previous round as well as with any comments made beforehand by peers. That data informed the decision-making of the last round [33].

While a higher number of rounds might have provided an even more robust set of results, the decision of moving ahead with the preset number of two rounds was made after a discussion within the research team that took into consideration practical and operational arguments. Namely, the burden of work requested from the experts across the different WPs and the timelines of the SAFEST project were crucial in the choice.

Patients' perspectives were highly valued from the research team's perspective. Consequently, their representation during the consensus-achieving process was ensured and specific steps were taken during the design of the umbrella review protocol in order to include Patient-Reported Experience Measures (PREMs) and Patient-Reported Outcome Measures (PROMs) in the ILM. We hypothesize that the anonymity of the feedback across rounds allowed for the neutralization of a potential dominance of the opinion of an experienced clinician or researcher against that from a patient or a patient representative.

Taking into consideration the results obtained, it is worth noting that certain crucial domains, such as quality of life, mental health, and satisfaction, were not assigned consensual priority during the process between both patients and professionals. To ensure that the CMS adequately reflects the needs and perspectives of patients, the research team is contemplating the identification of a patient-core set.

Nonetheless, some takeaways from the comparison of answers between patients and HCPs might be drawn. Of the top 5 highest rated measures by HCPs, none had an average rating below 8.4 by patients or patients' representatives. This is in line with the idea that patients may acquire some expertise in the area they experienced [34,35].

However, there are some numeric differences that our subgroup analysis captured and that highlight the need to involve patients and/or patients' representatives throughout the whole process of reshaping care towards improved quality and safety: from evidence generation to policymaking. Patients tend to value patient-relevant outcomes [36] higher in terms of importance than HCPs, which is corroborated by the results presented in Table 4.

On the other hand, HCPs value some crucial process measures higher, likely for being particularly aware of their impact on the provision of safe perisurgical care. This difference might be observed from the average importance ratings of measures such as "There is number of accredited healthcare professionals, including anaesthesiologists, surgeons and perioperative

**Table 4. Comparison of the average rating for importance of some patient-experience measures between patient or patient representatives and HCPs at the end of round two.**

| Indicator | HCPs average (standard deviation) | Patients average (standard deviation) |
|---|---|---|
| Pain associated fear | 6.49 (1.25) | 8.40 (0.85) |
| Pain (at rest or on movement) | 7.12 (1.07) | 8.25 (0.88) |
| Chronic pain | 6.43 (1.30) | 8.00 (0.95) |
| Time to lowest pain score | 6.21 (1.31) | 7.60 (1.27) |

specialized nurses that is considered adequate according to national recommendations" (Average patient rating: 6.40 / Average HCP rating: 7.72), "supervisory consultants are freely available to all junior anaesthesiologists" (Average patient rating: 6.80 / Average HCP rating: 7.49), or "The average case volume per surgeon is adequate according to national recommendations" (Average patient rating: 6.60 / Average HCP rating: 7.45).

Overall, the results of the present e-Delphi study paint a nuanced picture. While a substantial proportion of measures found common ground in terms of importance and feasibility, a considerable number failed to achieve consensus in both, reflecting the subtleties inherent to finding measures able to evaluate perioperative safety.

The overall higher average scores that almost every indicator received in terms of importance when compared to feasibility (Fig 1) stresses the crucial role that measurement plays when developing a set of measures.

Capturing the complexity of all the perioperative care needed and valued by patients, HCPs, and other key stakeholders is an intricate endeavor. The acknowledgment of this complexity is why measures were only considered for the FLM if they achieved consensus for both importance and feasibility.

## Conclusions

This study's findings contribute to the field of patient safety, specifically in the context of perioperative care. The perspective given by the results of this study will inform future concrete iterations of the SAFEST project and ultimately will support healthcare providers in the tracking, measurement, evaluation and regulation of their surgical activities.

In practice, so as to monitor and evaluate the implementation of the SAFEST strategy and recommendations in ten European hospitals, the consortium partners will curate an actionable subset of measures from the developed Core Measures Set. Guided self-evaluation and implementation materials will be made available to more than 100 hospitals in Europe and beyond.

To better grasp the complexity of perioperative safety measures, it is crucial to include a variety of perspectives from different groups in future studies on this topic. Notably, the results of the present study underline how critical the incorporation of patients and patients' representatives in health services research is. By involving these diverse perspectives, it is possible to achieve results that more comprehensively express actionable insights.

We hope this work can also help to pave the way for future research that should consolidate valid, co-created, patient-centered and action-oriented sets of measures on the topic of safe perisurgical care.

## Supporting information

**S1 Appendix. Final list of indicators.**
(DOCX)

**S1 Table. Detailed information on the characteristics of the experts.**
(DOCX)

## Acknowledgments

The authors would like to commend the SAFEST consortium members for their contribution: Daniel Arnal-Velasco, Joaquim Baneres, Ashish Bartakke, Hiske Calsbeek, Genis Carrasco, Pedro Casaca-Carvalho, Edoardo De Robertis, Yvette Emond, Neus Fabregas, Javier García-Silva, Pascal Garel, Oliver Groene, Anita Heideveld-Chevalking, Mari Kangasniemi, Janne Kommusaar, Kaja Kristensen, Andreia Leite, Irene Leon, Ismael Martínez-Nicolás, David

Marx, Marie Nabbe, Ana Beatriz Nunes, Carola Orrego, Kaja Polluste, Janne Pühvel, Eva Romero-Garcia, Yolanda Sanduende-Otero, Willemijn Schäfer, Caroline Schlinkert, Ayshe Seyfulayeva, Victor Soria-Aledo, Paulo Sousa, Joel Starkopf, Rosa Sunol, Helena Vall, Claudia Valli, Nina van der Schoot, Lilian Van Tuyl, Frantisek Vlcek, Marieke Voshaar, Cordula Wagner, Sophie Wang, Adam Žaludek, and Sandro Zamarian.

The authors would also like to acknowledge the SAFEST Scientific Advisory Group members for contributing to this study: Aamer Ahmed, Fragkiskos Angelis, Catarina Baptista, Metaxia Bareka, Kateryna Bielka, Mercedes Bilbao, Federico Bilotta, Elvira Bisbe, Dialina Brilhante, Pedro Carrascal, Pedro Delgado, Zsuzsanna Farkas-Pall, Loredana Gigli, Helen Haskell, Arvid Steinar Haugen, Jan Hofland, Beverley Hunt, Ib Jammer, Janek Kapper, Natasa Kovac, Susana Lorenzo, Rui Malheiro, Xose Manuel Meijome, Jannicke Mellin-Olsen, Margaret Murphy, Maria Ntalouka, Marta Dora Ornelas, Margarita Ovsepyan, Maria Papadakaki, Danica Rotar Pavlic, Julien Picard, Marek Pietruszka, Benedikt Preckel, Finn Radktke, José Manuel Rodríguez, Narimantas Evaldas Samalavicius, Pedro Vieira dos Santos, Ed Schoemaker, Kawaldip Sehmi, Henriëtte Smid-Nanninga, Joel Starkopf, John Tansley, Francesco Venneri, Matthias Weigl, and Argyro Zoumprouli.

## Author Contributions

**Conceptualization:** J. P. Dinis-Teixeira, Ana Beatriz Nunes, Andreia Leite, Willemijn L. A. Schäfer, Claudia Valli, Daniel Arnal-Velasco, Carola Orrego, Paulo Sousa.

**Data curation:** J. P. Dinis-Teixeira, Ana Beatriz Nunes.

**Formal analysis:** J. P. Dinis-Teixeira, Ana Beatriz Nunes.

**Investigation:** J. P. Dinis-Teixeira, Ana Beatriz Nunes, Andreia Leite, Willemijn L. A. Schäfer, Claudia Valli, Ismael Martínez-Nicolas, Ayshe Seyfulayeva, Pedro Casaca Carvalho, Anna Rodríguez, Irene Leon, Carola Orrego.

**Methodology:** J. P. Dinis-Teixeira, Ana Beatriz Nunes, Andreia Leite, Willemijn L. A. Schäfer, Claudia Valli, Ismael Martínez-Nicolas, Daniel Arnal-Velasco, Carola Orrego, Paulo Sousa.

**Project administration:** Ana Beatriz Nunes, Andreia Leite, Claudia Valli, Carola Orrego, Paulo Sousa.

**Supervision:** Carola Orrego, Paulo Sousa.

**Visualization:** J. P. Dinis-Teixeira, Ana Beatriz Nunes.

**Writing – original draft:** J. P. Dinis-Teixeira.

**Writing – review & editing:** J. P. Dinis-Teixeira, Ana Beatriz Nunes, Andreia Leite, Willemijn L. A. Schäfer, Claudia Valli, Ismael Martínez-Nicolas, Ayshe Seyfulayeva, Pedro Casaca Carvalho, Anna Rodríguez, Daniel Arnal-Velasco, Irene Leon, Carola Orrego, Paulo Sousa.

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
