## [Decision Letter · Decision Letter 0]

2 Aug 2024

PONE-D-24-16830Moving towards a core measures set for patient safety in perioperative care: an e-Delphi consensus studyPLOS ONE

Dear Dr. Dinis Teixeira,

Thank you for submitting your manuscript to PLOS ONE. After careful consideration, we feel that it has merit but does not fully meet PLOS ONE’s publication criteria as it currently stands. Therefore, we invite you to submit a revised version of the manuscript that addresses the points raised during the review process.

We look forward to receiving your revised manuscript.

Kind regards,

Bryan Kwun-Chung Cheng

Academic Editor

PLOS ONE

Journal Requirements:

"The work underlying this manuscript is encompassed in the European project SAFEST (Improving quality and patient SAFEty in surgical care through STandardisation and harmonisation of perioperative care in Europe).

This project receives funding from the European Union’s Horizon Europe research and innovation programme under grant agreement No 101057825.

Link: https://cordis.europa.eu/project/id/101057825"

Reviewers' comments:

Reviewer's Responses to Questions

**Comments to the Author**

1. Is the manuscript technically sound, and do the data support the conclusions?

Reviewer #1: Yes

2. Has the statistical analysis been performed appropriately and rigorously? 

Reviewer #1: Yes

3. Have the authors made all data underlying the findings in their manuscript fully available?

Reviewer #1: Yes

4. Is the manuscript presented in an intelligible fashion and written in standard English?

Reviewer #1: Yes

5. Review Comments to the Author

Reviewer #1: Authors:

1. The manuscript was well written and achieved your objective: "The objective of this study is to contribute to a consensus among experts, ultimately resulting in the formulation of a consensualized list of measures which might work as a stepping-stone for assessing perisurgical care safety and quality in healthcare." lines 127-129.

2. 48 of your 67 invited experts completed the two rounds. line 253 and 285-286.

3. Lines 374-378 addressed a limitation of systemic reviews.

4. You addressed the use of only two rounds of the Delphi process in this research.

5. "Taking into consideration the results obtained, it is worth noting that certain crucial domains, such

as quality of life, mental health, and satisfaction, were not assigned consensual priority during the

process between both patients and professionals." Lines 407-409. You addressed this. I think a positive/strength of this study was the inclusion of patients or patient advocacy voices a part of your experts.

6. I agree that your statement: "Overall, the results of the present e-Delphi study paint a nuanced picture." (line 432) Thus, you shared in detail and with explanations the structure of this particular Delphi process and you did not overstate claims.

7. Your conclusion seems appropriate: "This study’s findings contribute to the field of patient safety, specifically in the context of perioperative care." lines 445-446.

8. Your references are well entered and helpful. lines 481-576

9. Figures 1 and 2 were helpful demonstrations of your results, outcomes

10. Appendices S1 and S2 were noted and helpful.

6. PLOS authors have the option to publish the peer review history of their article (what does this mean?). If published, this will include your full peer review and any attached files.

Reviewer #1: No

---

## [Author Response · Author response to Decision Letter 0]

15 Sep 2024

In name of the authors, I would like to thank the reviewer for his/her thorough review of the article and such positive feedback on our work.

---

## [Editor Report · Decision Letter 1]

25 Sep 2024

Moving towards a core measures set for patient safety in perioperative care: an e-Delphi consensus study

PONE-D-24-16830R1

Dear Dr. Dinis Teixeira,

We’re pleased to inform you that your manuscript has been judged scientifically suitable for publication and will be formally accepted for publication once it meets all outstanding technical requirements.

Kind regards,

Bryan Kwun-Chung Cheng

Academic Editor

PLOS ONE
---

## [Editor Report · Acceptance letter]

14 Oct 2024

PONE-D-24-16830R1 

PLOS ONE

Dear Dr. Dinis Teixeira, 

I'm pleased to inform you that your manuscript has been deemed suitable for publication in PLOS ONE. Congratulations! Your manuscript is now being handed over to our production team.

Kind regards, 

on behalf of

Dr. Bryan Kwun-Chung Cheng 

Academic Editor

PLOS ONE